# Combining local model calibration with the emergent constraint approach to reduce uncertainty in the tropical land carbon cycle feedback

Nina Raoult[1], Tim Jupp[1], Ben Booth[2], and Peter Cox[1]

[1]College of Engineering, Mathematics and Physical Sciences, University of Exeter, Laver Building, North Park Road, Exeter, EX4 4QF, UK
[2]Met Office Hadley Centre, FitzRoy Road, Exeter, EX1 3PB, UK

**Correspondence:** Nina Raoult (n.m.raoult2@exeter.ac.uk)

**Abstract.** The role of the land carbon cycle in climate change remains highly uncertain. A key source of projection spread is related to the assumed response of photosynthesis to warming, especially in the tropics. The optimum temperature for photosynthesis determines whether warming positively or negatively impacts photosynthesis, thereby amplifying or suppressing $CO_2$ fertilisation of photosynthesis under $CO_2$-induced global warming. Land carbon cycle models have been extensively calibrated against local eddy flux measurements, but this has not previously been clearly translated into a reduced uncertainty in how the tropical land carbon sink will respond to warming. Using a previous parameter perturbation ensemble carried out with version 3 of the Hadley Centre coupled climate-carbon cycle model (HadCM3C), we identify an emergent relationship between the optimal temperature for photosynthesis, which is especially relevant in tropical forests, and the projected amount of atmospheric $CO_2$ at the end of the century. We combine this with a constraint on the optimum temperature for photosynthesis, derived from eddy-covariance measurements using the adjoint of the JULES land-surface model. Taken together, the emergent relationship from the coupled model and the constraint on the optimum temperature for photosynthesis define an emergent constraint on future atmospheric $CO_2$ in the HadCM3C coupled climate-carbon cycle under a common emissions scenario (A1B). The emergent constraint sharpens the probability density of simulated $CO_2$ change (2100-1900) and moves its peak to a lower value: $497 \pm 91$ compared to $607 \pm 128$ ppmv when using the equal-weight prior. Although this result is likely to be model and scenario dependent, it demonstrates the potential of combining the large-scale emergent constraint approach with parameter estimation using detailed local measurements.

## 1   Introduction

One of the key sources of uncertainty in future climate projections is the evolution of the land carbon sink (Friedlingstein et al., 2006; Cox et al., 2000; Arora et al., 2020; Canadell et al., 2021). As climate change elevates global temperatures and $CO_2$ conditions, the rate and efficiency of vegetation photosynthesis and respiration changes, influencing the capacity of the land to act as a repository for anthropogenic $CO_2$ (Medlyn et al., 1999; Cox et al., 2000; Friedlingstein et al., 2006). The structure and distribution of vegetation may also change in response to associate climate change, such as changes in precipitation patterns

(Trenberth, 2011). These responses provide a feedback on the initial climate change signal, potentially leading to key transitions and tipping points in the land biosphere. Notable examples include a global carbon sink to source transition, Amazon rainforest

dieback (Cox et al., 2004), shifting of the boreal forests (Chapin et al., 2004), and greening of the Sahel (Claussen et al., 2002).

Despite the increasing complexity of the climate-carbon cycle models developed for the latest IPCC (International Panel on Climate Change) Assessment Report (AR6), there is still a significant spread in projections of vegetation and soil carbon under common trajectories of atmospheric greenhouse gases and aerosols (Canadell et al., 2021). This spread arises partly from different climate projections within the host climate model and partly from uncertainties in the land surface models

themselves. Indeed, for the Joint U.K. Land Environment Simulator (JULES) land-surface model (Clark et al., 2011; Best et al., 2011) under one of the IPCC Special Report on Emissions Scenarios (SRES - A1B; Nakicenovic et al. (2000)), the atmospheric $CO_2$ change by the end of the century ($\Delta CO_2$) was found to range from 373.8 ppmv to 845.7 ppmv (Booth et al., 2012). This range was achieved simply by perturbing some of the model parameters related to the sensitivities of plant photosynthesis and soil respiration to temperature; stomatal conduction; soil water availability and surface evaporation; and

plant competition. The key source of projection spread was found to be related to the assumed response of photosynthesis to warming, especially in the tropics (Kattge and Knorr, 2007; Booth et al., 2012; Cox et al., 2013; Mercado et al., 2018). Indeed, the optimum temperature for photosynthesis ($T_{opt}$) is a common parameter in land-surface models that determines whether warming has a positive or negative impact on photosynthesis, thereby either amplifying or suppressing $CO_2$ fertilisation of photosynthesis under $CO_2$-induced global warming (Friedlingstein et al., 2006; Arora et al., 2020).

There is an urgent need to reduce such parametric uncertainties to make reliable and believable climate projections. Usually, to reduce uncertainty in model simulations, models are confronted with observations. However, although there is now an unprecedented amount of *in situ* and Earth Observation (EO) data with which to confront the models, the relatively shorter timescales mean these cannot be directly used to create constraints on changes in the Earth System over the next century. Furthermore, it is extremely computationally expensive to run complex land carbon cycle models (also known as land-surface

models - LSMs), within Earth System Models (ESMs) to produce multiple climate-carbon cycle projections. Instead, computationally efficient ways to translate short-term constraints into reductions in long-term projection uncertainty need to be developed.

Emergent constraints are used to bridge the gap between short-term contemporary observations and long-term future predictions (Cox et al., 2013; Wenzel et al., 2014, 2016; Hall et al., 2019; Williamson et al., 2021). Using the constraints provided

by observations and physical understanding available today, emergent constraints can be used to assess the relative likelihood of different long-term trends (Allen et al., 2002). Emergent constraints identified in the carbon cycle include the sensitivity of the annual growth rate of atmospheric $CO_2$ to tropical temperature anomalies (Cox et al., 2013), and the changing amplitude of the $CO_2$ seasonal cycle to the projected land photosynthesis (Wenzel et al., 2016; Hall et al., 2019; Williamson et al., 2021).

Data assimilation has been shown to be a useful and versatile tool to constrain the response of the carbon cycle in LSMs in the

short term. DA techniques use contemporary observations to improve the performance of a model by optimising two different components; either the values of unknown parameters (parameter estimation) or the predictions of the model according to a given data set (state estimation). In both cases, this is achieved by trying to find an optimal match between the model and

the observations by varying the properties of the model. In numerical weather prediction, DA has predominantly been used to optimise the state whilst keeping the parameters fixed. This is because the physics are mostly known and well-understood. However, in terrestrial carbon cycle models, where most of the equations are unknown, finding the correct set of parameters is more pertinent. These models can have over a hundred internal parameters representing the environmental sensitivities of the various land-surface and plant functional types. These parameters are generally chosen to represent measurable real-world quantities (e.g. surface albedo, plant root depth). This allows observationally-based estimates of these parameters to be made in the early stages of the model development process. However, the detailed performance of an LSM can be very sensitive to such internal parameters and so it is common for land-surface modellers to calibrate their models against available observations. Since optimisations give the best possible values of parameters given the model parameterisation and structural errors, the results are more reliable than field measurements of the same parameters, often taken a different spatial scales than model resolution.

In this study, we show how we can combine parameter optimisation with emergent constraint techniques to reduce uncertainty in future projections. Specifically, we derive an emergent constraint between a linear regression across the possible JULES $T_{opt}$ values between the change in $CO_2$ by the end of the century ($\Delta CO_2$), and the posterior distribution of parameter $T_{opt}$ optimised against GPP and LE *in situ* measurements.

## 2  Methods

### 2.1  A relationship between $T_{opt}$ and $\Delta CO_2$

In Booth et al. (2012)'s study, a large range of climate-carbon cycle feedbacks was found by perturbing the model parameters in the land surface component of the Hadley Centre global circulation model (version 3, HadCM3C). This experiment was conducted under the common climate scenario, A1B, which describes a future world of very rapid economic growth, a global population that peaks in the mid-century and declines after that, and the rapid introduction of new and more efficient technologies, with a balance of fossil intensive and non-fossil energy sources (Nakicenovic et al., 2000). One of the parameters perturbed in Booth et al. (2012) was $T_{opt}$, which corresponds to the optimal temperature for non-light limited photosynthesis for broadleaf forests. In JULES, non-light limited leaf-level photosynthesis is controlled by the carboxylation rate following the models of Collatz et al. (1991, 1992) with $T_{opt}$ representing the temperature at which the carboxylation rate reaches a maximum. This parameter was identified as the most important in controlling the carbon response of the model. Indeed, a statistically highly significant (p=0.000153) relationship between $T_{opt}$ and net $CO_2$ change by 2100 ($\Delta CO_2$) was found, whereas the rest of the parameters perturbed in the experiment showed little to no correlation with this change (Booth et al., 2012). $T_{opt}$ and $\Delta CO_2$ were shown to be anti-correlated, with higher values of $T_{opt}$ resulting in lower values of $\Delta CO_2$. This implies that when the optimal temperature for photosynthesis for broadleaf trees is high, more $CO_2$ is predicted to be removed from the atmosphere through increased $CO_2$ fertilisation. This is particularly relevant in the tropics, where in a warming world, ambient temperatures have the potential to exceed optimal photosynthetic temperature persistently, and where broadleaf trees represent large carbon stocks (Booth et al., 2012).

Using linear regression, we can exploit this relationship to calculate a probability distribution function (PDF) for the distribution of $\Delta CO_2$ given $T_{opt}$, i.e., $P\{\Delta CO_2|T_{opt}\}$. The contours of equal probability density around the best-fit linear regression follow a Gaussian probability density

$$P\{\Delta CO_2|T_{opt}\} = \frac{1}{\sqrt{2\pi\sigma_f^2}}\exp\left\{-\frac{(\Delta CO_2 - f(T_{opt}))^2}{2\sigma_f^2}\right\} \tag{1}$$

where $f$ is the function describing the linear regression between $\Delta CO_2$ and $T_{opt}$, and $\sigma_f$ is the "prediction error" of the regression.

## 2.2  A constraint on $T_{opt}$ using local eddy-flux measurements

The land-surface component of HadCM3C was the Met Office Surface Exchange Scheme (MOSES, Cox et al. (1999)), which became the Joint U.K. Land Environment Simulator (JULES). The adJULES system (Raoult et al., 2016) was developed

specifically to optimise the internal parameter of the JULES land surface model using data assimilation. Data assimilation allows the integration of multiple types of data ($\mathbf{y}$) in order to optimise model parameters ($\mathbf{x}$) while making allowance for associated uncertainties. It is a powerful tool which allows for objective and repeatable calibrations. A Bayesian framework is used to include prior knowledge about the parameters ($\mathbf{x}_b$). All errors are assumed to be Gaussian distributed (with $\mathbf{R}$ and $\mathbf{B}$ the prior error covariance matrices for the observations and parameters, respectively). The optimisation corresponds to minimising

the mismatch ($J$) between the model outputs and the observed data with respect to $\mathbf{x}$:

$$J(\mathbf{x}) = \frac{1}{2}[(\mathbf{y} - M(\mathbf{x}))^T\mathbf{R}^{-1}(\mathbf{y} - M(\mathbf{x})) - (\mathbf{x} - \mathbf{x}_b)^T\mathbf{B}^{-1}(\mathbf{x} - \mathbf{x}_b)] \tag{2}$$

where $M(\mathbf{x})$ is the model output vector given $\mathbf{x}$. Methods for minimising the cost function range from stochastic random search algorithms to deterministic gradient-based methods.

This second class of methods was integrated into the adJULES system (Raoult et al., 2016). The adJULES system uses

the adjoint of the JULES model, a computationally efficient way used to calculate the gradient of Eq. 2. The adjoint allows for efficient and repeatable optimisations utilising the gradient information. The quasi-Newton algorithm L-BFGS-B (limited memory Broyden–Fletcher–Goldfarb–Shanno algorithm with bound constraints; see Byrd et al., 1995) is used to minimise the cost function iteratively. At each iteration of the algorithm, the cost function and its gradient with respect to each parameter are evaluated. The adjoint also allows for the accurate calculation of the Hessian (second derivative of the cost function) at the

optimum. The Hessian determines the posterior error covariance matrix, which is used to calculate the posterior uncertainties associated with the best-fit parameters (in the form of PDFs).

Deriving the adjoint of a model as complex as JULES is extremely costly. Fortunately, this has been done for JULES v2.2, which uses the same photosynthesis model as MOSES, allowing us to optimise the same parameters and photosynthesis model as used in HadCM3C and, therefore, in Booth et al. (2012)'s perturbation experiment. In Raoult et al. (2016), adJULES was

used to improve the model performance at a wide range of broadleaf sites by optimising the key land surface parameters perturbed in Booth et al. (2012). Each parameter was assigned a wide prior distribution, allowing the parameters to take values

from a large range of credible values elicited from expert opinion. The optimisation was performed using monthly *in situ* gross primary productivity (GPP) and latent heat (LE) data from $CO_2$ eddy-fluxes measured at FluxNet sites (Baldocchi et al., 2001; Pastorello et al., 2020). The FluxNet database contains more than 500 locations worldwide, and all of the data are processed in a harmonised manner using the standard methodologies including correction, gap-filling, and partitioning (Papale et al., 2006). A large number of broadleaf sites were selected from this database (27 in total; see Raoult et al. (2016) for details). The optimisation was performed in multi-site configuration, i.e., simultaneously over all selected sites, as well as over both fluxes, to find a single set of best-fit model parameters and their associated uncertainties. The optimisation returned best-fit parameters with posterior distributions much narrower than the prior, reducing the range of viable parameter values. From these posterior distributions, we obtain an observational-constrained PDF for $T_{opt}$, i.e., $P(T_{opt})$.

## 2.3 Calculation of the PDF for $\Delta CO_2$

We follow the method used by Cox et al. (2018) to bring these two elements together and calculate the PDF for $\Delta CO_2$. The PDF for $\Delta CO_2$ is calculated by numerically integrating over the product of two PDFs, $P\{\Delta CO_2|T_{opt}\}$ and $P(T_{opt})$:

$$P(\Delta CO_2) = \int\limits_{-\infty}^{\infty} P\{\Delta CO_2|T_{opt}\}P(T_{opt})dT_{opt}. \tag{3}$$

## 3 Results and discussion

Figure 1 shows how the distribution of likely $T_{opt}$ values, i.e., $P(T_{opt})$, changes when the JULES LSM is optimised against local measurements of photosynthesis (GPP) and latent heat (LE) using the adJULES system (Raoult et al., 2016). We can see that the posterior distribution is much more pronounced than the prior and suggests a higher parameter value than previously used. Values of $T_{opt}$ taken from this distribution, when used in the JULES model, will result in the best fit of the model to local measurements of photosynthesis (GPP) and latent heat (LE), and therefore improve the model's credibility.

As well as displaying the results found by optimising simultaneously over all of the broadleaf sites found in Raoult et al. (2016), Fig. 1 also considers distributions of $T_{opt}$ found optimising at each individual broadleaf site. Though none of these gives such a narrow distribution, the majority do suggest that the optimal value for the parameter (shown by the peak of the distributions) is higher than previously used in the JULES model. This gives confidence in the posterior distribution found by calibrating over all sites. Furthermore, one of the known limitations of gradient-based methods is their tendency to get stuck in local minima (i.e., not finding the 'true' global minimum). Optimisations over multiple sites have been shown to be more robust, with the additional constraints from each site acting to smooth the cost function, thus making local minima less common. As such, multi-site optimisations are more reliable in finding the true best-fit parameters and associated PDFs. For the remainder of this study, we will solely use the posterior distribution found by calibrating over all sites.

Through this multi-site calibration, we find $T_{opt}$ (i.e., the optimal temperature for non-limited photosynthesis for broadleaf forests) to be around 35°C with an uncertainty of approximately $\pm0.9$°C. This value falls well within the typical 30-40°C temperature range observed in most leaf-scale photosynthetic-temperature response curves (Kattge and Knorr, 2007). However,

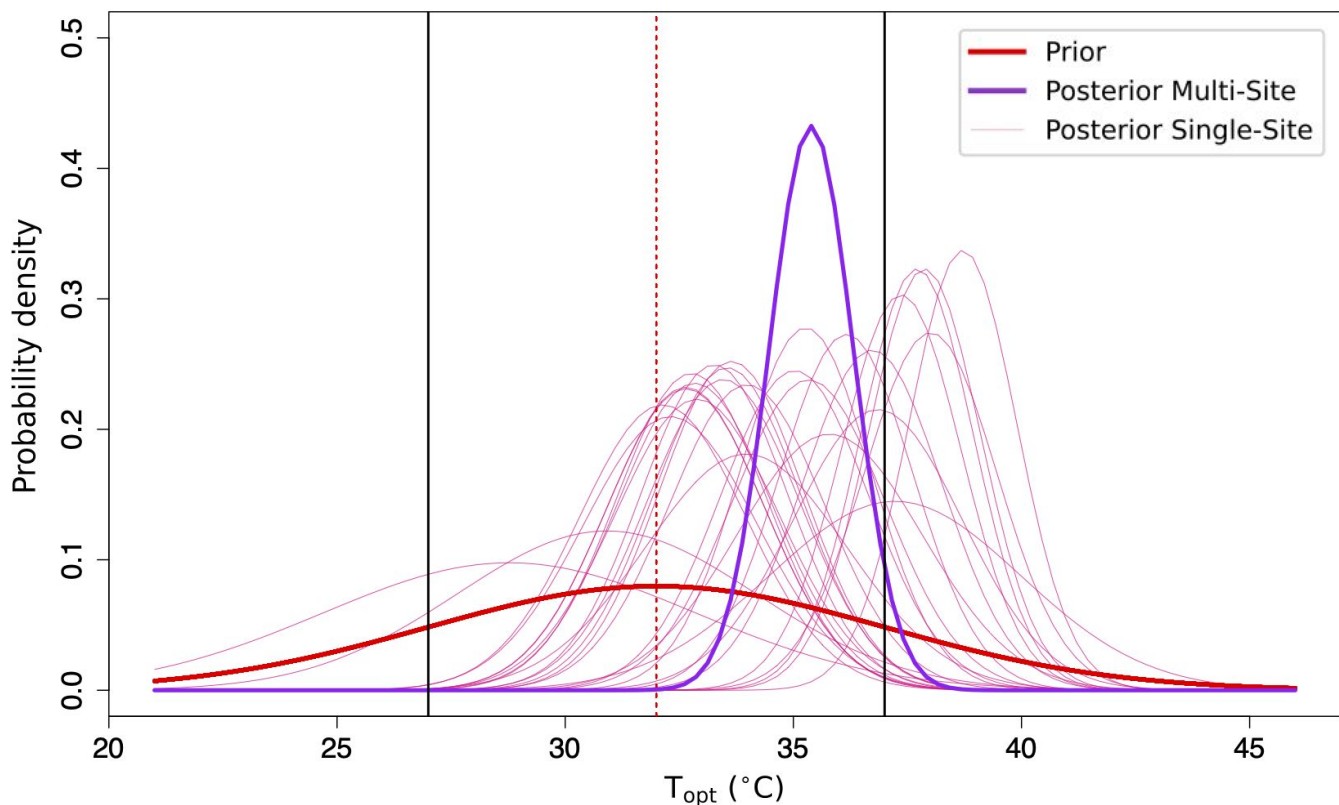

**Figure 1.** Different PDFs of $P(T_{opt})$ found when using the adJULES system to optimise the JULES land-surface model against Fluxnet data. The prior distribution (red) of the parameter is compared to the posterior distribution (purple) found by calibrating simultaneously over the 27 broadleaf FluxNet sites considered in Raoult et al. (2016) (i.e., a multi-site optimisation), as well as the individual posterior distributions found by calibrating at each site separately (i.e., single-site optimisations). All distributions are modelled by a Gaussian curve. Note the range used in optimisation (entire $x$-axis) is greater than the range used in Booth et al. (2012) (vertical black lines). Initial value of $T_{opt}$ in JULES is highlighted by the dashed red line.

land surface models are not commonly run at leaf-scale - especially not when run within wider Earth System Models to predict climate change. Furthermore, $T_{opt}$ at leaf-scale has been shown to differ from $T_{opt}$ at ecosystem level (Field et al., 1995; Huang
155 et al., 2019), where additional processes limiting photosynthesis may be impacted by temperature changes (e.g., accelerated leaf ageing at high atmospheric temperatures). While Huang et al. (2019) showed that the global mean of $T_{opt}$ at ecosystem scale was lower ($23 \pm 6°C$) than leaf-level values - they also showed that warmer regions had higher ecosystem scale $T_{opt}$ values than cooler ones. Indeed for the tropics - these values were found to be close to growing-season air temperature, and again similar to the values of $T_{opt}$ obtained through the adJULES calibration.
160     We can now translate the reduction the uncertainty in the $T_{opt}$ into a reduction of uncertainty in carbon-climate feedbacks. Instead of running computationally expensive climate models with a new set of the parameter ensembles generated from the

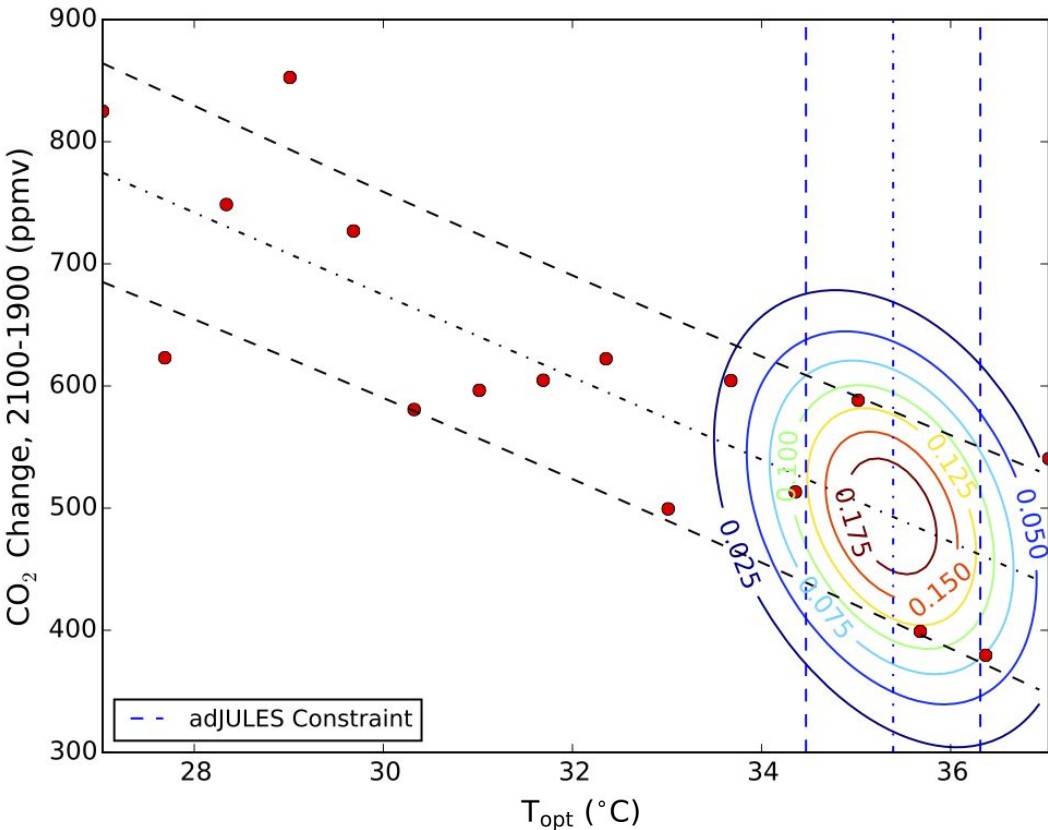

**Figure 2.** Contours of probability density for the linear regression adapted from Booth et al. (2012). The red dots show the relationship between different $T_{opt}$ values and the resulting change in $CO_2$ by the end of the century from the parameter perturbation experiment of Booth et al. (2012) (see Table A1 of these values). The thin black dashed-line shows the best-fit linear regression, and the thick black lines show plus and minus the prediction error (see Methods). The vertical blue lines show the observational constraint on $T_{opt}$ value, with the best fit shown by the thin dashed blue line, and the thick vertical dashed lines showing plus and minus one standard error about this value. The continuous contours are the product of these two underlying PDFs. The integral of these contours across the $x$-axis variable leads to the $T_{opt}$-constrained PDF shown in Figure 3a.

posterior distribution, this posterior PDF in $T_{opt}$ can be directly translated into a PDF for atmospheric carbon change using the carbon cycle sensitivity identified in Booth et al. (2012) as an emergent constraint. The linear relationship between $T_{opt}$ and $CO_2$ change is shown in Fig. 2. The vertical blue lines included in this figure show the $T_{opt}$ constraint from adJULES.
165 These lines are found at the upper-end of the figure and select a narrow range of $T_{opt}$ values. Using this constraint, we can derive tighter bounds on the $CO_2$ response of the model. The linear regression and $T_{opt}$ constraint can then be used to generate contours from the product of the two PDFs, and hence the $T_{opt}$-constrained PDF of $CO_2$ change between 1900 and 2100.

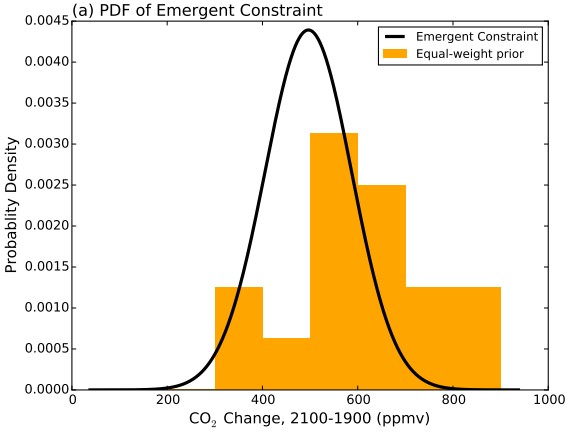

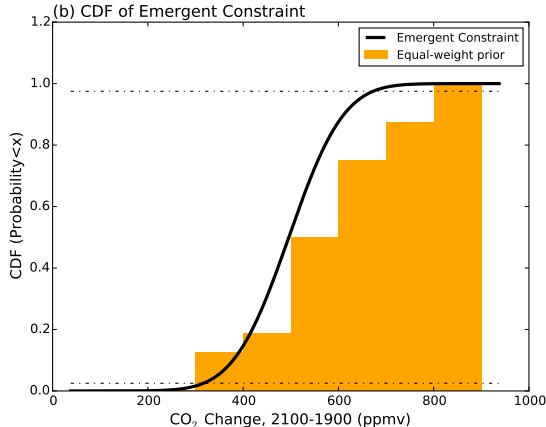

(a) The probability density histogram for the unconstrained $T_{opt}$ values (orange) and the conditional PDF arising from the emergent constraint (black).

(b) The cumulative distribution for the unconstrained $T_{opt}$ values (orange) and the conditional CDF arising from the emergent constraint (black).

**Figure 3.** Emergent constraint on the sensitivity of $T_{opt}$ to the magnitude of future carbon cycle response. The horizontal dot-dashed lines show the 95% confidence limits on the CDF plot. The orange histograms (both panels) show the prior distributions that arise from equal weighting of parameter perturbation experiment in 500 ppmv bins.

Figure 3a shows this PDF. This PDF is compared to the histogram arising from assuming that all of the $T_{opt}$ values in the ensemble are equally likely to be true. The emergent constraint from the $T_{opt}$ optimisation sharpens the PDF of $CO_2$ change (2100-1900) and moves its peak to a lower values: $496.5 \pm 91$ compared to $606.6 \pm 128$ ppmv when using the equal-weight prior. Figure 3a shows the resulting cumulative density function (CDF), which gives the probability of $CO_2$ change (2100-1900) taking a value lower than the value shown on the $x$-axis. The 95% confidence limits (shown by the black horizontal lines) range from 300 ppmv to 650 pmv. We see that values higher than 650 ppmv become extremely unlikely. The $T_{opt}$ constraint, therefore, reduces the estimated probability of $CO_2$ change values, predicting a slightly stronger carbon sink over broadleaf trees than previously suggested by the JULES climate predictions and reducing the range of possible responses by 30% and discounting higher values of $CO_2$ change. Although both the calibration of $T_{opt}$ (Raoult et al., 2016) and the parameter perturbation experiment were conducted globally (Booth et al., 2012), the latter found that the dominant cause of the spread in future $CO_2$ was due to the tropical land, and specifically due to the assumed optimum temperature for photosynthesis tropical forests.

## 4   Conclusions

Data assimilation and emergent constraints are two powerful techniques which can enable more precise projections of climate change. By bridging the gap between both techniques, we have shown that optimisations can be used not only to improve the

current state of the model but also to constrain climate predictions. Short-scale half-hourly observations spanning only a few years can be used to inform us about expected changes in the next century. By severely reducing the uncertainty in $T_{opt}$, we have reduced the uncertainty in the $CO_2$ change predicted by JULES under HadCM3C for the end of the century under the A1B climate scenario. These results are no doubt model and scenario dependent. Nevertheless, this study highlights a new methodology to use should future models show strong emergent relationships between model parameters and future climate change.

JULES is a complex and ever-evolving land-surface model, with more processes regularly added. Newer versions JULES now exist, so an updated parameter perturbation experiment would need to be conducted to understand the new sensitivities of the model to future climate change. However, running JULES coupled with a climate-carbon model like HadGEM3 (the successor to HadCM3C) to test these sensitivities requires a lot of time and resources. Instead, we may need to rely on different tools such as the IMOGEN (Integrated Model Of Global Effects of climatic aNomalies, Huntingford et al. (2010)) system, an emulator of climate change using pattern-scaling. Furthermore, developing the adjoint of the newest version of JULES is complicated. Deriving the adjoint of complex models like JULES is costly and becomes quickly outdated as the model versions advance. Fortunately, newer optimisation schemes have become available (e.g., LAVENDAR, Pinnington et al. (2020)), which still allows for posterior PDFs to be generated after each optimisation.

This study acts as a proof of concept, a blueprint for constraining future projections of a land-surface model. We have shown that observational datasets are crucial in helping us understand and reduce uncertainty large-scale climate feedback. With the growing amount of observational data available, both from *in situ* and satellite, there is a unique opportunity to perform multiple data stream optimisations, increasing the credibility of the posterior parameter distributions. There are many data sets we could use to constrain the carbon cycle, including the inter-annual variability of leaf area index, solar-induced fluorescence and atmospheric $CO_2$. Furthermore, due to the strong coupling between the carbon-water-energy cycles, we could move to use other constraints to optimise the model parameters, such as soil moisture and land surface temperature. Note that unlike the more orthodox application of DA in weather forecasting, the Raoult et al. (2016) study used DA for parameter estimation to derive optimum JULES parameters to fit FluxNet observational data rather than to nudge state variables. The paper shows that the resulting constraint on the optimum temperature for photosynthesis ($T_{opt}$) in turn provides an emergent constraint on the increase in atmospheric CO2 by 2100 in a coupled climate-carbon cycle model (Booth et al., 2012). Although this clear link is very likely to be model dependent, we present it here as a first example of how local model calibration and the emergent constraint technique can be used to constrain global climate-carbon cycle projections.

*Code and data availability.* The code and data used in this paper are available in the following online repository: https://github.com/NRaoult/ adJULES

**Table A1.** Results from Booth et al. (2012)'s parameter perturbation experiment.

| $T_{opt}$ | $CO_2$ change (2100-1900) |
|---|---|
| 31.0000 | 589.773 |
| 33.6667 | 597.978 |
| 36.3333 | 373.826 |
| 35.6667 | 392.767 |
| 27.6667 | 616.082 |
| 31.6667 | 599.082 |
| 27.0000 | 817.798 |
| 29.0000 | 845.664 |
| 29.6667 | 720.893 |
| 33.0000 | 492.605 |
| 32.3333 | 616.539 |
| 37.0000 | 534.170 |
| 34.3333 | 506.310 |
| 28.3333 | 742.913 |
| 30.3333 | 573.196 |
| 35.0000 | 581.707 |

*Author contributions.* NR and PC designed the study. BB provided data from the parameter perturbation experiment. NR and TJ build the adJULES system used to constrain JULES parameters. NR and PC generated the figures. All authors contributed to writing and editing the text.

*Competing interests.* The authors declare no competing interests.

*Acknowledgements.* This work has received funding from the European Union's Horizon 2020 research and innovation programme under the Marie Skłodowska-Curie grant agreement No 101020076.

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
