# Peer review of "Combining local model calibration with the emergent constraint approach to reduce uncertainty in the tropical land carbon cycle feedback"

_EGUsphere, 2023_

## Referee Comment (RC2)

The accurate projections of future climate change impacts on land surface carbon cycles are key to understand the climate change carbon cycle feedback and to mitigate climate change. This paper provides a way by using the observational-constraints of the optimal temperature and the emergent relationship between optimal temperature and atmospheric $CO_2$ changes to narrow the uncertainty in the projected future $CO_2$ changes. This method combined the short-term optimization with the long-term climate-carbon feedback and provided a new way of understanding the climate change.

I enjoyed reading the manucript in its novel idea. While before it can be accepted for publication, I have some questions on its suitability for application to broader model groups.

1. This study used the relationship between $T_{opt}$ and atmospheric $CO_2$ changes, over the tropics for the broadleaf forests. I was wondering about the atmospheric $CO_2$ used for the global mean or the tropical regions? Since the global $CO_2$ can also be mediated by other vegetation types.

2. This study used the adjoint of JULES, which happened to be of the land component of the Earth system model that is used for projections. I wonder how can this relationship be transferred to other models, such as the CMIP5/6 models?

3. Data assimilation is a good tool for optimizing parameters from different processes. The nonlinearity of the terrestrial ecosystem models can have few parameters that are interacted and this would result in the joint-distributions of parameters from different processes. While in the data assimilation we seldom considered that or put little focus on the parameter interactions. So how can we properly obtain the relationships between paramters and variables that can be projected to futures? As the authors mentioned soil moisture and other variables. Why do not we use the emergent relationships between optimized variables instead?

---

## Author Response (AR1)

**Reviewer 1**

The authors have identified an emergent relationship between the optimum temperature of photosynthesis and the projected change in atmospheric CO2 between 2100 and 1900, using a combination of global climate-carbon cycle modeling and local eddy-covariance measurements. Results of the analysis show that the larger Topt could further generated a lower $\triangle$CO2 at the end of the century than the original model predictions. Overall this is a well-written and solid study. The findings are also of broad interest to the community and offer an important constrain on the magnitude of the carbon cycle feedback. I have just a few questions about the data processing procedure and would like to see more discussion about the Topt in the manuscript.

We would like to thank the reviewer for taking the time to read and comment on the manuscript - the added discussion around Topt will definitely strengthen the manuscript.

Please specify the meaning of the red dots in figure 2, which will help readers who have not read Booth et al (2012) to have a clearer understanding of the emergent constrain in your manuscript, at least providing the details of the simulations (red dots) in the appendix.

The red dot in Figure 2 shows the results from Booth et al., i.e., the relationship between different Topt values and the resulting change in $CO_2$ by the end of the century. We have expanded the caption of the figure to include this description and added the table of Topt vs delta $CO_2$ adapted from Booth et al (2012) to the appendix.
> "Contours of probability density for the linear regression adapted from Booth et al., 2012. **The red dots show the relationship between different Topt values and the resulting change in $CO_2$ by the end of the century from the parameter perturbation experiment of Booth et al., (2012) (see appendix for a table of these values).** The thin black dashed-line shows the best-fit linear regression...."

I suggest the authors better explain the concept and calculation of Topt as well as the optimization process. Also, please provide more details about the utilization of the GPP and LE data in the analysis.

We have added the following text to better explain the concept of $T_{opt}$ in our study (L81):
> One of the parameters perturbed in Booth et al., (2012) was $T_{opt}$, which corresponds to the optimal temperature for non-light limited photosynthesis for broadleaf forests. **In JULES, non-light limited leaf-level photosynthesis is controlled by the carboxylation rate following the models of Collatz et al. (1991, 1992) with $T_{opt}$ representing the temperature at which the carboxylation rate reaches a maximum.**

We have also added more on $T_{opt}$ in the discussion (see the following response).

The optimisation is currently described in Section 2.2. We have expanded as follows:

> **The optimisation was performed in multi-site configuration, i.e., simultaneously over all selected sites, as well as over both fluxes, to find a single set of best-fit model parameters and their associated uncertainties.**

As for the FluxNet data, these are only used in the calibration step. We have expanded their description as follows (L121):

> The optimisation was performed using monthly in situ gross primary productivity (GPP) and latent heat (LE) data from $CO_2$ eddy-fluxes measured at FluxNet sites (Baldocchi et al., 2001; Pastorello et al., 2020). **The FluxNet database contains more than 500 locations worldwide, and all of the data are processed in a harmonised manner using the standard methodologies including correction, gap-filling, and partitioning (Papale et al., 2006). A large number of broadleaf sites were selected from this database (27 in total; see Raoult et al., (2016) for details).**

Since the author derived Topt using the adjoint of the JULES land-surface model and local eddy flux measurements, I suggest adding a paragraph to look a little more deeply at the result of Topt (see additional paper below).

We thank the reviewer for this suggestion. As well as adding extra detail on Topt to the methods (in response to a previous comment), we have added the following uo results section:

> **"Through this calibration, we find $T_{opt}$ (i.e., the optimal temperature for non-limited photosynthesis for broadleaf forests) to be around 35°C with an uncertainty of approximately ± 0.9°C. This value falls well within the typical 30-40°C temperature range observed in most leaf-scale photosynthetic-temperature response curves (Kattge, J. & Knorr, 2007). However, land surface models are not commonly run at leaf-scale - especially not when run within wider Earth System Models to predict climate change. Furthermore, $T_{opt}$ at leaf-scale has been shown to differ from $T_{opt}$ at ecosystem level (Field et al., 1995; Huang et al., 2019), where additional processes limiting photosynthesis may be impacted by temperature changes (e.g., accelerated leaf ageing at high atmospheric temperatures). While Huang et al., 2019 showed that the global mean of Topt at ecosystem scale was lower (23 ± 6°C) than leaf-level values - they also showed that warmer regions had higher ecosystem scale Topt values than cooler ones. Indeed for the tropics - these values were found to be close to growing-season air temperature, and again similar to the values of Topt obtained through the adJULES calibration."**

Huang et al., Air temperature optima of vegetation productivity across global biomes. Nat. Ecol. Evol. 3, 772–779 (2019)

**Reviewer 2**

The accurate projections of future climate change impacts on land surface carbon cycles are key to understand the climate change carbon cycle feedback and to mitigate climate change. This paper provides a way by using the observational constraints of the optimal temperature and the emergent relationship between optimal temperature and atmospheric CO2 changes to narrow the uncertainty in the projected future CO2 changes. This method combined the short-term optimization with the long-term climate-carbon feedback and provided a new way of understanding the climate change.

I enjoyed reading the manuscript in its novel idea.

Thank you, we are glad the reviewer enjoyed reading our manuscript and thank them for providing insightful comments.

While before it can be accepted for publication, I have some questions on its suitability for application to broader model groups.

1. This study used the relationship between Topt and atmospheric CO2 changes, over the tropics for the broadleaf forests. I was wondering about the atmospheric CO2 used for the global mean or the tropical regions? Since the global CO2 can also be mediated by other vegetation types.

Atmospheric $CO_2$ changes are due to the prescribed fossil fuel emissions, and the global ocean and land carbon sinks. As the reviewer correctly suggests, the latter is due to the total response across the land-surface, rather than just in the tropics. However, in the carbon cycle ensemble results of Booth et al. (2012) the dominant cause of spread in future $CO_2$ was due to the tropical land, and specifically due to the assumed optimum temperature for tropical forests. We have added the following to the discussion (L160):

**Although both the calibration of $T_{opt}$ (Raoult et al., 2016) and the parameter perturbation experiment were conducted globally (Booth et al., 2012), the latter found that the dominant cause of the spread in future $CO_2$ was due to the tropical land, and specifically due to the assumed optimum temperature for photosynthesis tropical forests.**

2. This study used the adjoint of JULES, which happened to be of the land component of the Earth system model that is used for projections. I wonder how can this relationship be transferred to other models, such as the CMIP5/6 models?

Interesting point. We believe this is an artefact of the UK ESM CMIP5 model. However, is it possible that similar relationships exist in other and more recent models, although we would need to perform costly parameters perturbation experiments to unearth them. We have added the following to the conclusions to highlight this (L167):

**These results are no doubt model and scenario dependent. Nevertheless, this study highlights a new methodology to use should future models show strong emergent relationships between model parameters and future climate change.**

3. Data assimilation is a good tool for optimizing parameters from different processes. The nonlinearity of the terrestrial ecosystem models can have few parameters that are interacted and this would result in the joint-distributions of parameters from different processes. While in the data assimilation we seldom considered that or put little focus on the parameter interactions. So how can we properly obtain the relationships between paramters and variables that can be projected to futures? As the authors mentioned soil moisture and other variables. Why do not we use the emergent relationships between optimized variables instead?

Our apologies, it seems that we may not have been clear enough about the nature of the Data Assimilation in this study, which is used to constrain internal model parameters rather than state variables. We will add the following text to clarify this:

"we could move to **use other constraints to optimise the model parameters**, such as soil moisture and land surface temperature. **Note that Unlike the more orthodox application of Data Assimilation (DA) in weather forecasting, the Raoult et al. (2016) study used DA to derive optimum JULES parameters to fit FluxNet observational data, rather than to nudge state variables. The paper shows that the resulting constraint on the optimum temperature for photosynthesis ($T_{opt}$) in turn provides an emergent constraint on the increase in atmospheric $CO_2$ by 2100 in a coupled climate-carbon cycle model (Booth e al., 2012). Although this clear link is very likely to be model dependent, we present it here as a first example of how local model calibration and the emergent constraint technique can be used to constrain global climate-carbon cycle projections.**"